# Impact of Tyrosine Kinase Inhibitors on the Immune Response to SARS-CoV-2 Vaccination in Patients with Non-Small Cell Lung Cancer

**DOI:** 10.3390/vaccines11101612

**Published:** 2023-10-19

**Authors:** Norma Hernández-Pedro, Marisol Arroyo-Hernández, Pedro Barrios-Bernal, Eunice Romero-Nuñez, Victor A. Sosa-Hernandez, Santiago Ávila-Ríos, José Luis Maravillas-Montero, Rogelio Pérez-Padilla, Diego de Miguel-Perez, Christian Rolfo, Oscar Arrieta

**Affiliations:** 1Laboratorio de Medicina Personalizada, Instituto Nacional de Cancerología, S.S.A., San Fernando 22 Sección XVI, Tlalpan, Mexico City 14080, Mexico; nhernandezp@incan.edu.mx (N.H.-P.); pedrobarrios@ciencias.unam.mx (P.B.-B.); eunice.romero@gmail.com (E.R.-N.); 2Thoracic Oncology Unit, Instituto Nacional de Cancerología, S.S.A., San Fernando 22 Sección XVI, Tlalpan, Mexico City 14080, Mexico; marisol.neumologia@gmail.com; 3Red de Apoyo a la Investigación, Instituto Nacional de Ciencias Médicas y Nutrición Salvador Zubirán, Mexico City 14080, Mexico; md.victor.sosa@gmail.com (V.A.S.-H.); maravillas@cic.unam.mx (J.L.M.-M.); 4Centro de Investigación en Enfermedades Infecciosas (CIENI), Instituto Nacional de Enfermedades Respiratorias, Calzada de Tlalpan 4502, Belisario Domínguez Sección XVI, Tlalpan, Mexico City 14080, Mexico; santiago.avila@cieni.org.mx; 5Department of Research on Tobacco and COPD, Instituto Nacional de Enfermedades Respiratorias, Calzada de Tlalpan 4502, Belisario Domínguez Sección XVI, Tlalpan, Mexico City 14080, Mexico; perezpad@unam.mx; 6Mount Sinai Health System, Icahn School of Medicine at Mount Sinai, New York, NY 11776, USA; diego.demiguelperez@mssm.edu (D.d.M.-P.); christian.rolfo@mssm.edu (C.R.)

**Keywords:** tyrosine kinase inhibitors, lung cancer, SARS-CoV-2, COVID-19 vaccines, antigen-secreting cells, B-lymphocytes

## Abstract

Immune dysregulation and cancer treatment may affect SARS-CoV-2 vaccination protection. Antibody production by B-cells play a vital role in the control and clearance of the SARS-CoV-2 virus. This study prospectively explores B-cell seroconversion following SARS-CoV-2 immunization in healthy individuals and non-small cell lung cancer (NSCLC) patients undergoing oncological treatment. 92 NSCLC patients and 27 healthy individuals’ blood samples were collected after receiving any COVID-19 vaccine. Serum and mononuclear cells were isolated, and a serum surrogate virus neutralization test kit evaluated SARS-CoV-2 antibodies. B-cell subpopulations on mononuclear cells were characterized by flow cytometry. Patients were compared based on vaccination specifications and target mutation oncological treatment. A higher percentage of healthy individuals developed more SARS-CoV-2 neutralizing antibodies than NSCLC patients (63% vs. 54.3%; *p* = 0.03). NSCLC patients receiving chemotherapy (CTX) or tyrosine kinase inhibitors (TKIs) developed antibodies in 45.2% and 53.7%, of cases, respectively, showing an impaired antibody generation. CTX patients exhibited trends towards lower median antibody production than TKIs (1.0, IQR 83 vs. 38.23, IQR 89.22; *p* = 0.069). Patients receiving immunotherapy did not generate antibodies. A sub-analysis revealed that those with ALK mutations exhibited non-significant trends towards higher antibody titers (63.02, IQR 76.58 vs. 21.78, IQR 93.5; *p* = 0.1742) and B-cells quantification (10.80, IQR 7.52 vs. 7.22, IQR 3.32; *p* = 0.1382) against the SARS-CoV-2 spike protein than EGFR patients; nonetheless, these differences were not statistically significant. This study shows that antibodies against SARS-CoV-2 may be impaired in patients with NSCLC secondary to EGFR-targeted TKIs compared to ALK-directed treatment.

## 1. Introduction

Lung cancer (LC) patients are vulnerable to severe infections of the coronavirus disease (COVID-19). For instance, a retrospective analysis of 1524 patients in Wuhan, China revealed a higher susceptibility to COVID-19 in non-small cell lung cancer (NSCLC) patients (OR = 2.31) compared to the general population [1]. Moreover, the TERAVOLT global LC registry reported a mortality rate of approximately 30% for LC patients hospitalized for SARS-CoV-2 virus infection in 2020 [2]. Nonetheless, it was noted that COVID-19 vaccination reduced mortality and hospitalization risk in patients with thoracic neoplasms and COVID-19, and this effect was enhanced using an additional booster (OR = 0.30, *p* = 0.0003) [1]. This suggests that COVID-19-derived immunity affects a patient’s prognosis in cancer patients.

However, LC patients are characterized by a disturbed immunity derived from SARS-CoV-2 vaccination [3,4]. For example, an observational study conducted in Japan reported lower SARS-CoV-2 seroconversion in LC patients versus control individuals (96.7% vs. 100%; *p* < 0.001) [3]. Likewise, findings from a UK national study of COVID-19 identified undetectable levels of anti-S antibodies in most cancer patients compared with controls [4]. The reasoning behind these findings pointed to the immunomodulatory role of oncological treatment. Chemotherapy and radiotherapy are widely known to affect immunological response against SARS-CoV-2 [5]. 

As such, patients receiving chemotherapy (CTX) or targeted therapy harbored lower immunoglobulin G (IgG) levels against spike protein of SARS-CoV-2 than those receiving immunotherapy following vaccination with BNT162b2 (BioNTech; Pfizer) (OR = 5.4; 95% CI, 1.5–20.2; *p* = 0.02) [6]. Similarly, CTX patients had lower nucleocapsid protein IgG levels than those without it [6]. Similarly, a retrospective study of cancer patients who underwent testing for IgG against SARS-CoV-2 demonstrated higher titers of antibodies after immunotherapy than with anti-CD-20 or stem cell transplant [7]. In this context, a third dose has been recommended to boost the immune response in patients undergoing cancer treatment, as a study evidenced higher frequency of serological response was registered after three doses of the COVID-19 vaccine compared to only two doses in 163 cancer patients (75% vs. 65%) [8]. 

Nonetheless, little is known about the influence of tyrosine kinase inhibitors (TKIs) in immune responses derived from COVID-19 vaccines in NSCLC patients with EGFR and ALK alterations. The most similar approaches to this issue are sub-group analyses from larger studies showing that TKI treatment is associated with a reduced antibody response to the BNT162b2 vaccine in LC patients compared to healthy controls [9,10]. 

Moreover, as immunity against SARS-CoV-2 is not limited to seroconversion, some reports have shown that B cells signatures harbor prognostic importance in non-cancer patients diagnosed with severe COVID-19, demonstrating that decreases in memory B cells and increments in antibody-secreting cells and CD19^+^ B cells are positively related to the severity of this disease [11]. As an extrapolation of these findings, B-cells subsets have also been studied in individuals with hematologic malignancies and COVID-19, showing that mortality in these patients was closely related to defects in CD4^+^ and B-cells quantifications. Consequently, individuals recovering from COVID-19 were those able to exhibit a SARS-CoV-2-specific CD4 and CD8 T cell response, along with subsequent increases in antibody titers and memory B cells against infection. Thus, diverse lymphocyte sub-populations are essential in cancer immune response against SARS-CoV-2. This association remains unexplored in LC patients.

Available studies on the role of target therapy in immunity to SARS-CoV-2 vaccination are usually small studies that focus on one type of vaccine and do not describe B-cell activity after vaccination. This study examined the seroconversion rate and B-cell signature in lung adenocarcinoma patients after SARS-CoV-2 vaccination.

## 2. Materials and Methods

### 2.1. Study Design and Participants

This prospective longitudinal study of two cohorts was conducted at the Instituto Nacional de Cancerología in Mexico (INCan) from September 2021 to December 2021. This study was conducted in accordance with the Declaration of Helsinki, and the protocol was approved by the ethics and research committees of INCan (approval number: 022/006/ICI; CEI/1586/21). Informed consent was obtained from all participants. 

During this period, participants were vaccinated against SARS-CoV-2 according to the national vaccination program applied to the Mexican population. Participants received mRNA vaccines, such as BNT162b29 (Pfizer-BioNTech, NY, USA) and Johnson & Johnson’s Janssen, viral vector vaccines including ChAdOx1 (Oxford/AstraZeneca, Cambridge, UK), Gam-COVID-Vac (Sputnik V) and Ad5-nCoV-S (CanSino), or inactivated virus vaccines like Sinovac-CoronaVac in one or two doses according to the pre-established scheme. Vaccines were administered according to those available at the local healthcare centers.

The first cohort comprised LC patients over 18 years recruited at their routine visit to the oncology service. All cancer patients received CTX, immunotherapy, tyrosine kinase inhibitors (TKIs) or concomitant treatments when having the first and second doses of COVID-19 vaccines. The second cohort consisted of healthy individuals without a known cancer diagnosis or other disorders leading to immunodeficiency. 

All participants were asked for vaccination status against COVID-19 and previous history of SARS-CoV-2 infection. Vaccine adverse effects were documented according to the Mexican official standard for epidemiological surveillance, which grades the severity of symptoms in four groups in increasing order. 

Clinical data regarding cancer treatment were obtained from medical records. Blood samples were collected 30 days after the second dose of the COVID-19 vaccine to determine antibodies against SARS-CoV-2 spike protein (S protein), CD19^+^ B-cells, antibody-secreting B cells (ASBC), CD27(−) B cells, and memory B lymphocytes.

We excluded participants with clinical suspicion or microbiological evidence of active COVID-19 infection. 

### 2.2. Determination of Neutralizing Antibodies Using ELISA

Blood samples were collected (6 mL) in an EDTA tube, then centrifuged at 2000 rpm for 15 min with 2 mL of Cytiva Ficoll-Paque (Thermo Fisher Scientific, Inc., Waltham, MA, USA. # 11768538) to isolate serum and mononuclear cells from other blood components. Afterward, serum was stored in cryotubes at −85 °C until antibody analysis. Meanwhile, mononuclear cells were washed using Phosphate-buffered saline (PBS, 1X) (Gibco, Thermo Fisher Scientific, Waltham, MA, USA. # 10010023) and resuspended in a 500 µL solution of 90% fetal bovine serum (Gibco, Thermo Fisher Scientific, Waltham, MA, USA. # 2614007) and 10% dimethyl sulfoxide (Thermo Fisher Scientific, Inc., Waltham, MA, USA. # 10127403), which was finally frozen at −85 °C until flow cytometry analysis. 

Determination of SARS-CoV-2 antibodies was performed using a surrogate virus neutralization test kit (Cell Science, Newburyport, MA, USA # CKV001). In a 96-well plate, we added 50 µL dilutions of positive and negative controls and samples per well. Then, the plate underwent 30 min of incubation at 37 °C. Afterward, each well was complemented with 50 mL of detection solution A, mixed for 5 min with a microplate mixer and incubated for 1 h at 37 °C. 

After incubation, the used reagent was discarded from the wells and washed thrice with 300 µL of Wash Buffer. Then, 100 µL of TMB color reagent was added to each well and incubated for 10 min at 37 °C. Finally, it was added 50 µL of stop solution to each well. After 10 min, we determined the optical density of each well using a microplate reader set to 450 nanometers with a wavelength correction of 540–570 nanometers. Cut-offs to determine positivity or negativity for SARS-CoV-2 neutralizing antibodies were calculated using the following formula: Inhibition = [1−OD value of sample/OD value of negative control)] × 100%. As a result, OD450 values of ≥20% indicate positivity to neutralizing antibodies, while <20% was considered as negativity.

### 2.3. Evaluation of B Cell Populations by Flow Cytometry

Cells were defrizzed, resuspended in RPMI-1640 media with phenol red (Gibco, Thermo Fisher Scientific, Waltham, MA, USA. #114529-5G) and counted with a TC20 automated cell counter (Bio-Rad, Hercules, CA, USA. #145-0101). Peripheral blood mononuclear cell (PBMCs) staining was performed using conjugated monoclonal antibodies; dead cells were excluded using live/dead discrimination by Zombie Green Fixable Viability Kit staining (BioLegend, CA, USA. #423111), as well as singlets discrimination with size and complexity parameters before cell analysis. Cells were treated with a Human TruStain FcX (BioLegend, San Diego, CA, USA. #422301) for 10 min for extracellular staining. 

Cells were incubated for 30 min at 4 °C with an antibody cocktail: Anti-CD19 APC (BioLegend, San Diego, CA, USA #302212), Anti-CD27 PE (BioLegend, San Diego, CA, USA #302808) and Anti-CD38 FITC (BioLegend, San Diego, Ca, USA #303504). Cells were centrifuged at 1500 rpm for 5 min and fixed with Fixation Buffer (BioLegend, San Diego, CA, USA #420801). Finally, cells were washed once with cell staining buffer (BioLegend, San Diego, CA, USA #420201) and then resuspended in 500 μL of buffer for immediate (no more than a 12 h delay) flow cytometric analysis on a BD LSR Fortessa Cell Analyzer (BD Biosciences, San Jose, CA, USA) using BD FACSDiva v9.0 software (BD Biosciences, San Jose, CA, USA). Up to 1 × 10^6^ cells were analyzed using FlowJo v10 software (BD Biosciences, USA) and developed using Fluorescence Minus One (FMO) control to define gates. Compensation was assessed using Compensation Beads (BD Biosciences, San Jose, CA, USA #552843) and single-stained F fluorescent samples.

Analysis and cell markers are described in Appendix A.

### 2.4. Statistical Analysis

The type of distribution of data was calculated by the Kolmogorov-Smirnov test. As all measured parameters represented a non-normal distribution, they were presented as medians and interquartile ranges (IQR). The Mann-Whitney U test was used for the inter-group comparison of antibody titers and B-cells, and the Chi-square test was used for determining differences among clinical characteristics. Statistical analysis was performed in 26.0 version SPSS (IBM, New York, NY, USA), and graphs were created using GraphPad software (Version 7, San Diego, CA, USA).

## 3. Results

### 3.1. Clinical Characteristics

Samples from ninety-two patients with thoracic neoplasms and 27 healthy subjects were analyzed to determine SARS-CoV-2 antibodies. The median sampling time was 34 days after the second dose of the COVID-19 vaccine. Among all patients, 58.8% (n = 70) were female, 41.2% (n = 49) were male, and their median age was 59 years (IQR 30–81) (Table 1). In the NSCLC cohort, 92.1% (n = 82) of patients had IV-stage disease, and only 7.9% (n = 7) were early-stage. Adenocarcinoma was the most common type of cancer in the overall sample 85.9% (n = 79), while mesothelioma 5.4% (n = 5), squamous cell carcinoma 4.3% (n = 4), and other neoplasms 4.3% (n = 4) were found in a smaller proportion of cases. This study analyzed a sample of 79 patients diagnosed with adenocarcinoma to examine the impact of treatment on the production of antibodies against SARS-CoV-2. The most frequent histological grade for adenocarcinomas was intermediate (34.2%, n = 27), followed by high (29.1%, n = 23), low (19%, n = 15) and NE (17.7%, n = 14) grades. The proportion of patients receiving TKIs was 71.4% (n = 55), CTX was 23.4% (n = 18), immunotherapy was 3.9% (n = 3), radiotherapy was 1.3% (n = 1), and two patients were treatment naïve. Only 10 patients had a previous COVID-19 infection (10.9%) (Table 2). 

### 3.2. SARS-CoV-2 Vaccines Distribution 

In the healthy cohort, 36% (n = 9) of subjects received Oxford/AstraZeneca (AZD1222), 28% (n = 7) Pfizer-BioNTech (BNT162b2), 20% (n = 5) Johnson & Johnson’s Janssen, 12% (n = 3) Sputnik V and 4% (n = 1) Sinovac. In the meantime, NSCLC patients received AZD1222 in 39.6% (n = 36), BNT162b2 in 34.1% (n = 31), Sinovac in 14.3% (n = 13), Sputnik V in 11% (n = 10) and CanSino in only 1.1% (n = 1) of cases (Table 1). 

### 3.3. Vaccine-Related Adverse Effects

The most prevalent vaccine-related adverse effects in the healthy cohort were fatigue grade ≥ 1 (28%, n = 7), pain grade ≥ 1 (20%, n = 5) and arthralgias grade ≥ 1 (16%, n = 4). In contrast, the NSCLC group experienced fatigue (20.7%, n = 19), pain (19.6%, n = 18), headache (17.4%, n = 16) and arthralgias (10.9%, n = 10) ≥ 1. These adverse effects were unrelated to SARS-CoV-2 antibodies (Table 3).

### 3.4. Vaccine-Derived Immune Response against SARS-CoV-2 among Healthy Individuals and NSCLC Patients

Positivity for SARS-CoV-2 neutralizing antibodies was defined as values over 20% of inhibition. According to this, participants younger than 59 years old produced more antibodies than those older than 59 years (65.9% vs. 43.8%; *p* = 0.033), harboring a trend to higher titers of antibody production (61% vs. 43.1%; *p* = 0.098), but without statistical significance (Table 2). As shown in Figure 1A, a greater proportion of subjects younger than 59 years produced antibodies against spike proteins (median 73.8; IQR 84.73) than those older than 59 (median 13.02; IQR 75.35; *p* = 0.0002). A similar pattern of immune response was exhibited by lung cancer patients, showing that (age < 59 years: median 58.51; IQR 89.64 vs. age > 59 years: median 15.22; IQR 83.06; *p* = 0.0228) (Figure 1B). Among the lung cancer subgroup, TKI-treated subjects showed no significant differences in antibody production according to age (< 59 years: median 46.12; IQR 79.29 vs. age > 59 years: 93.42; IQR 58.45; *p* = 0.1369) (Figure 1C). There was a discernible disparity in the levels of neutralizing antibodies among cancer patients when stratified by age, although this factor did not represent a significant modifier of immune response among those TKI treatment (mean = 59 years)**.**

In addition, patients with LC had lower antibody levels (33.26, IQR 94.18) than those in the control group (49.10, IQR 87.93; *p* = 0.0316), as shown in Figure 2A. Furthermore, the subgroup analysis revealed that there were no statistically significant disparities in CD19^+^ B-cell levels between NSCLC patients (7.59; IQR 3.43) and healthy individuals (6.61; IQR 5.25; *p* = 0.6859). However, patients with NSCLC had a lower number of antibody-secreting cells (2.26; IQR 3.48) than healthy people (3.84; IQR 6.72; *p* = 0.04) (Figure 2B). A trend toward a greater amount of CD19^+^ B-cells was also identified among NSCLC subjects having CD27(−) cells (75.55; IQR 19.38) compared to healthy individuals with CD27(−) cells (67.50; IQR 18.6; *p* = 0.0747). Similarly, no statistical differences were seen in memory cell quantification between lung cancer (20.10; IQR 17.07) and healthy groups (27.8; IQR 14.15; *p* = 0.2304) (Figure 2C). Thus, NSCLC patients exhibited lower antibody production, likely derived from lower quantifications of antigen-secreting cells (ASC). On the contrary, CD27-negative cells and memory cells did not display important disparities among both groups. 

### 3.5. The Immune Response Induced by AZD1222 or BNT162b2 Vaccines between Healthy or NSCLC Patients

After characterizing serological immune response in both healthy and lung cancer groups, a comprehensive sub-analysis was performed comparing serological responses derived from the most prevalent vaccines (AZD1222 and BNT162b2) administered in lung cancer patients and healthy subjects. In the lung cancer group, AZD1222 induced lower antibody titers against the S protein (54.0; IQR 96.49 vs. 99.04; IQR 14.53; *p* = 0.004) and lower levels of memory B cells (16.90; IQR 17.7 vs. 29.60; IQR 15.43; *p* = 0.055) compared with the control group. Nevertheless, antibody-secreting cells (3.91; IQR 5.93 vs. 2.94; IQR 2.73), CD19^+^ (1.94; IQR 2.84 vs. 3.48; 3.67) or CD27(−) (79.20; IQR 19.60 vs. 67.0; IQR 14.40) lymphocytes showed no significant differences between groups after vaccination with AZD1222 (Appendix A). They were determined non-statistically significant higher BNT162b2-induced antibody titers in individuals with NSCLC than in the healthy group (33.04; IQR 82.04 vs. 14.67; IQR 89.44). No differences were found in CD19^+^ B cells (7.46; IQR 3.06 vs. 6.61; IQR 5.29), antibody-secreting cells (3.48; IQR 3.67 vs. 1.90 IQR 2.67), CD27(−) cells (73.90; IQR 20.38 vs. 78.30; IQR 21.45) and memory lymphocytes (16.90; IQR 20.56 vs. 21.40; IQR 17.45) between NSCLC patients and healthy subjects (Appendix A). Therefore, relevant disparities in immune response were observed between groups regarding antibody titers and memory B cells only in individuals who underwent vaccination with AZD1222. 

### 3.6. Immunogenicity Derived from COVID-19 Vaccines in Patients Receiving CTX or TKIs

Patients on CTX had decreased antibody titers against S proteins (1.02; IQR 83.06 vs. 38.23; IQR 87.49; *p* = 0.0465). No statistical differences were detected in the B-cell levels (6.95; IQR 2.34 vs. 7.71; IQR 6.39; *p* = 0.2466), antibody-secreting cells (3.20; IQR 6.72 vs. 2.10; IQR 2.97; *p* = 0.2685) and CD27(−) cells (79.20; IQR 11.60 vs. 73.90 IQR 19.08; *p* = 0.2357). There was a downward trend in memory B lymphocytes (13.90; IQR 8.6 vs. 21.40; IQR 17.65; *p* = 0.0873) compared with those undertaking EGFR TKIs.

(Figure 3A–C). Altogether, chemotherapy may affect antibody production and memory B cells quantifications compared with TKIs. On the contrary, B, CD27(−) and antigen-secreting cells did not display relevant changes between treatments.

### 3.7. Immunological Consequences of Vaccines in NSCLC Patients Undergoing Targeted Therapy against EGFR Mutations or ALK Alterations

After analyzing immune-response according to each therapeutic modality, a sub-group analysis among those undergoing TKI therapy revealed important immunological disparities among those with EGFR or ALK alterations. Strikingly, ALK-mutated patients undergoing therapy with TKIs showed non-significant increases in antibody titers against S proteins (ALK: 63.02; IQR 77.58 vs. EGFR: 21.78; IQR 93.59; *p* = 0.1742) and B-cells quantification (ALK: 10.80; IQR 7.52 vs. EGFR: 7.22; IQR 3.32; *p* = 0.1382) (Figure 4A,B) compared to those with EGFR mutations. Nonetheless, there were no significant differences in terms of antibody-secreting cells (ALK: 2.66; IQR 8.83 vs. EGFR: 2.28; IQR 2.67; *p* = 0.1540), CD27(−) cells (ALK: 72.50; IQR 16.90 vs. EGFR: 80.0; IQR 20.20; *p* = 0.3565) and memory B cells (ALK: 20.50; IQR 14.9 vs. EGFR: 16.90; IQR 20.8; *p* = 0.4662) (Figure 4C) between patients with ALK or EGFR mutations. Patients with mutant ALK factor did not affect neutralized antibody production (70.8 vs. 29.2; *p* = 0.059). In detail, the production of SARS-CoV-2 neutralizing antibodies was evaluated according to the type of therapy they received: 45.2% (14/31) of patients on CTX developed antibodies, and 53.7% (29/55) of patients on TKIs developed antibodies. According to the median antibody production, patients on CTX had lower antibodies (1.0, IQR 83) than patients receiving TKIs (38.23; IQR 89.22; *p* = 0.069). Only three patients received immunotherapy and did not develop antibodies. The EGFR group exhibited a lower level of antibody production in comparison to the ALK group (21.78 vs. 63.02, *p* = 0.1742). Patients over 59 years old and EGFR mutations present fewer antibodies (9.4 vs. 95.8; *p* = 0.02). It is noteworthy that individuals below 59 years old and with ALK mutations exhibited a lower production of antibodies compared to other groups (54.0 vs. 86.4; *p* = 0.44) (Table 2). Therefore, EGFR TKIs affect serological immune response, likely inhibiting spike-directed antibody titers, CD19^+^ B cells and antigen-secreting cells (ASC). On the contrary, non-significant differences were identified according to mutational status regarding CD27(−) and memory B cells. 

## 4. Discussion

This study explored the impact of antineoplastic treatment on seroconversion derived from COVID-19 vaccination in patients with non-small cell lung cancer compared to a healthy control group. Healthy patients presented higher antibody titers than those with NSCLC. Even so, the rates in the control group were lower than those reported in other studies. Seroconversion of antibodies against SARS-CoV-2 was up to 90% in healthy individuals and >94% in patients with solid malignancies. In contrast, in our study, only 60% of the control group and 52% of patients with NSCLC were seroconverted [10,12]. LC patients exhibited lower antibody titers for SARS-CoV-2 than in previous studies, probably secondary to factors involved in antineoplastic therapy. Previous evidence described that cancer patients undergoing CTX exhibit diminished antibody production following vaccination. According to this idea, NSCLC patients showed lower antibody titers in those treated with CTX than with TKIs. Still, this phenomenon is mainly attributable to those with ALK alterations, as patients undergoing targeted therapy against EGFR mutations were associated with lower antibody quantifications than those with ALK alterations. The biological reasoning behind this phenomenon is still poorly understood, but in vitro, reports highlight the importance of EGFR in the activity of immune cells, as EGFR-derived signaling is closely related to reduced activation in Th2 lymphocytes [13]; therefore, we theorize that EGFR inhibition is likely affecting antibody synthesis against SARS-CoV-2. The most similar approach to this phenomenon is the study of Lavallade et al. [14], which showed that TKIs against BCR-ALB translocation affect B-cell immune responses in chronic myeloid leukemia.

Furthermore, lower antibody production in NSCLC patients compared to healthy individuals is likely derived from diminished quantifications of antigen-secreting cells, as these are important agents in serological response [15]. Moreover, we measured CD19^+^ and CD27(−) lymphocytes, and antibody-secreting cells, as these are closely related to antibody production and harbor a prognostic role in COVID-19 patients [11,16]. This study identified trends in lower quantifications of B-lymphocytes and antibody-secreting cells in NSCLC patients undergoing TKIs for EGFR mutations compared to those with ALK alterations. This further supports that this targeted therapy may impair B-lymphocytes´ function in NSCLC patients. Unfortunately, little is known about the effects of EGFR TKIs on these cells in NSCLC. Still, some studies have demonstrated that B cell activity harbors a dynamic character, as it is modifiable by external interventions. Andreano et al. [15] showed that a third dose of vaccine increases the antibody neutralization potency against all variants of SARS-CoV-2 compared to that derived from a second dose, primarily due to the expansion of new B-cells, which were not detected after primary immunization. This may explain why we did not find significant increases in B-cell quantifications in our cohort, as a third dose of the SARS-CoV-2 vaccine is needed to enhance their activation and expansion. Another example of this phenomenon is found in T-cells of patients with hematologic malignancies, demonstrating that BNT162b2 induced activation and persistence of memory T cells up to six months post-vaccination [17]. Therefore, immunological response in our NSCLC cohort may be affected by a lack of a third dose of the SARS-CoV-2 vaccine and the use of targeted therapy against EGFR mutations. However, it is essential to note that studies on this relationship are limited. Finally, the main limitations of this study are its small sample size and a lack of a third dose of the COVID-19 vaccine due to heterogeneity in the availability of COVID-19 vaccinations among the general population. Our results could also be affected by certain differences in terms of age between both the healthy and the lung cancer groups. Thus, more rigorous research is needed to determine the impact of TKIs on the oncological immune response against SARS-CoV-2. 

## 5. Conclusions

Reduced antibody titers against SARS-CoV-2 and trends to lower levels of antigen-secreting cells were identified in NSCLC patients with EGFR mutations who received TKIs compared to those undergoing ALK-directed therapy. Thus, EGFR-targeted therapy may affect antibody production and B-cell activity in NSCLC.

## Figures and Tables

**Figure 1 vaccines-11-01612-f001:**
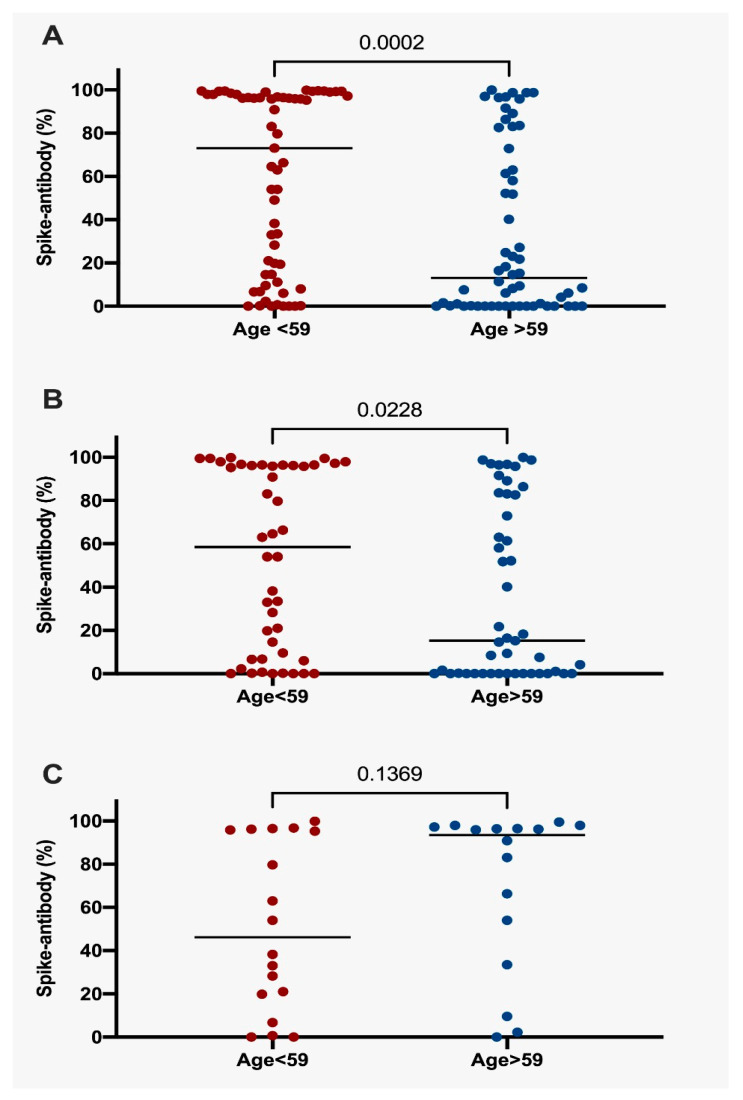
Percentage of spike-antibodies (**A**) Percentage of spike-antibodies in all participants according to median age. (**B**) Percentage of spike-antibodies in NSCLC patients according to median age. (**C**) Percentage of spike-antibodies in NSCLC patients with TKI treatment. Statistical significance: *p* value ≤ 0.05. NSCLC, Non-Small Cell Lung Cancer.

**Figure 2 vaccines-11-01612-f002:**
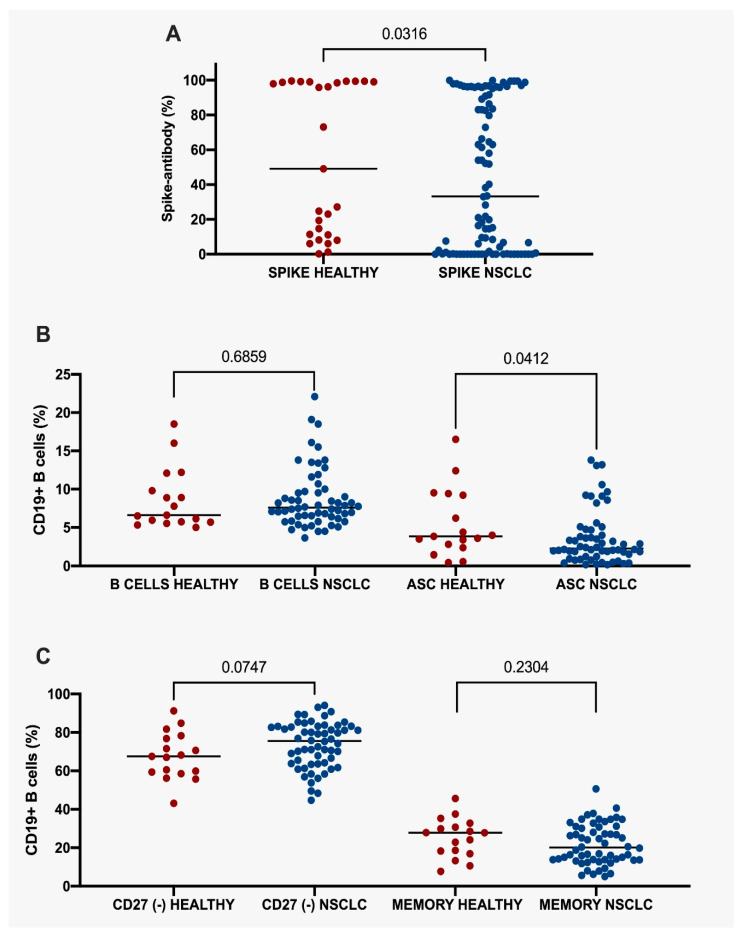
Percentage of spike-antibodies between healthy and NSCLC groups. (**A**) Percentage of spike-antibodies between all healthy and NSCLC patients. (**B**) Percentage of CD19^+^ B cells and ASC between healthy and NSCLC groups. (**C**) CD27(−) and memory CD19^+^ B cells between healthy and NSCLC groups. Statistical significance: *p* value ≤ 0.05. NSCLC, Non-Small Cell Lung Cancer. ASC, Antibody-Secreting Cells.

**Figure 3 vaccines-11-01612-f003:**
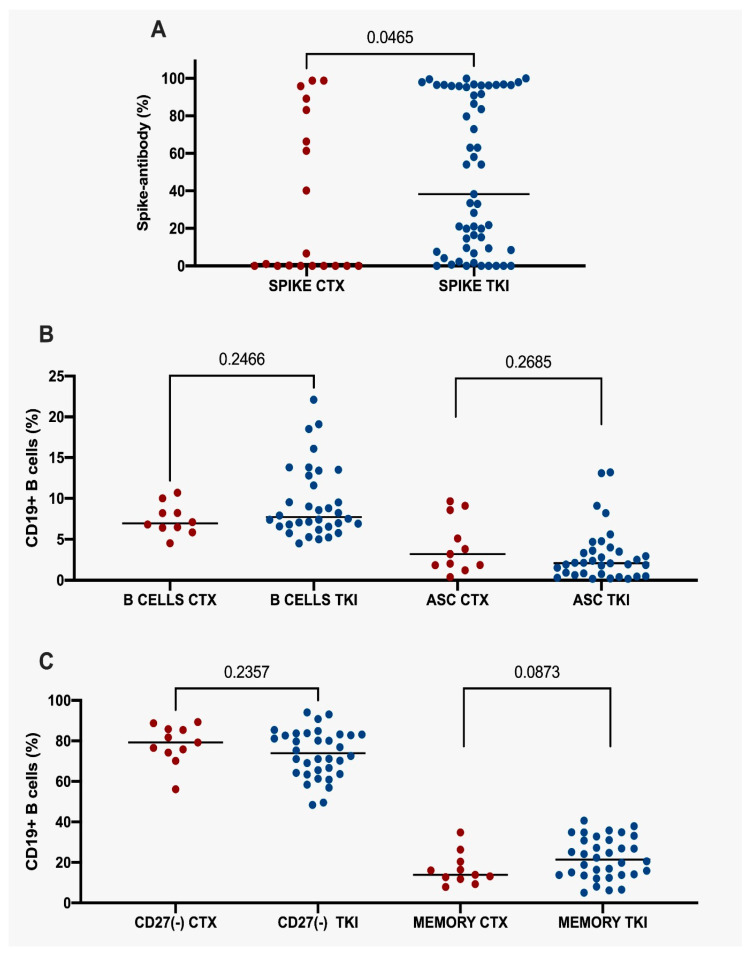
Percentage of spike-antibodies in NSCLC undergoing chemotherapy or tyrosine kinase inhibitors. (**A**). Percentage of spike-antibodies in all NSCLC undergoing chemotherapy or tyrosine kinase inhibitors. (**B**) Percentage of CD19^+^ B cells and ASC in NSCLC undergoing chemotherapy or tyrosine kinase inhibitors. (**C**) CD27(−) and memory CD19^+^ B cells in NSCLC undergoing chemotherapy or tyrosine kinase inhibitors. Statistical significance: *p*-value ≤ 0.05. NSCLC, Non-Small Cell Lung Cancer. ASC, Antibody-Secreting Cells. CTX, chemotherapy. TKI, tyrosine kinase inhibitors.

**Figure 4 vaccines-11-01612-f004:**
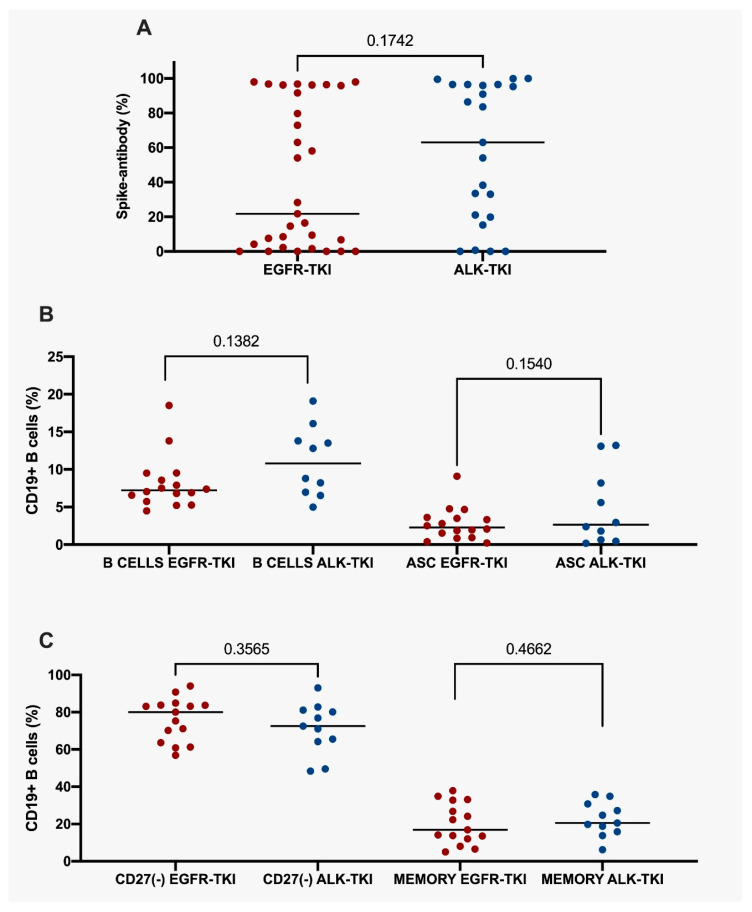
Percentage of spike-antibodies in NSCLC harboring EGFR mutations or ALK alterations undergoing tyrosine kinase inhibitors. (**A**) percentage of spike-antibodies in all NSCLC harboring EGFR mutations or ALK alterations undergoing tyrosine kinase inhibitors. (**B**) percentage of CD19^+^ B cells and ASC in NSCLC harboring EGFR mutations or ALK alterations undergoing tyrosine kinase inhibitors. (**C**) CD27(−) and memory CD19^+^ B cells in NSCLC harboring EGFR mutations or ALK alterations undergoing tyrosine kinase inhibitors. Statistical significance: *p* value ≤ 0.05. NSCLC, Non-Small Cell Lung Cancer. ASC, Antibody-Secreting Cells. TKI, tyrosine kinase inhibitors. EGFR, Epidermal Growth Factor Receptor. ALK, anaplastic lymphoma kinase.

**Table 1 vaccines-11-01612-t001:** Clinical features of the general population, including healthy individuals and NSCLC patients.

Clinical Characteristics N = 119	Healthy % (n)	NSCLC % (n)	*p* Value
Gender			0.663
Male: 41.2 (49)	20.4 (10)	79.6 (39)	
Female: 58.8 (70)	24.3 (17)	75.7 (53)	
Age			0.179
<59: 47.9 (57)	28.1 (16)	71.9 (41)	
≥59: 52.1 (62)	17.7 (11)	82.3 (51)	
COVID-19			0.808
Yes: 10.9 (13)	23.1 (3)	76.9 (10)	
No: 89. 1 (106)	22.6 (24)	77.4 (82)	
SARS-CoV-2 vaccine			0.002
BNT162b: 33.6 (40)	22.5 (9)	77.5 (31)	
AZD1222: 38.7 (46)	19.6 (9)	80.4 (37)	
Sputnik V: 10.9 (13)	23.1 (3)	76.9 (10)	
Sinovac: 11.8 (14)	7.1 (1)	92.9 (13)	
Johnson & Johnson’s Janssen: 4.2 (5)	100 (5)	0 (0)	
CanSino: 0.8 (1)	0 (0)	100 (1)	

NSCLC, Non-Small Cell Lung Cancer. SARS-CoV-2, severe acute respiratory syndrome coronavirus 2. COVID-19, coronavirus disease of 2019. BNT162b2, BioNTech-Pfizer vaccine. AZD1222, Oxford-Astra Zeneca vaccine.

**Table 2 vaccines-11-01612-t002:** Clinical features of NSCLC patients stratified by SARS CoV-2 antigen load and titles.

N = 92	SARS-CoV-2 Antigen (<20)% (n)	SARS-CoV-2 Antigen (>20)% (n)	*p* Value	SARS-CoV-2Titles (−)% (n)	SARS-CoV-2Titles (+)% (n)	*p* Value
Gender						
Male: 39	46.2 (18)	53.8 (21)		46.2 (18)	53.8 (21)	
Female: 53	45.3 (24)	54.7 (29)	0.934	50.9 (27)	49.1 (53)	0.678
Age						
<59: 41	34.1 (15)	65.9 (29)		39 (16)	61 (25)	
≥59: 51	56.3 (27)	43.8 (21)	0.033	56.9 (29)	43.1 (22)	0.098
Smoking						
No: 67	41.8 (28)	58.2 (39)		46.3 (31)	53.7 (36)	
Yes: 25	56.0 (14)	44.0 (11)		56 (14)	44 (1)	0.485
Woodsmoke Exposure						
No: 72	48.6 (35)	51.4 (37)		50 (36)	50 (36)	
Yes: 20	35 (7)	65 (13)	0.280	45 (9)	55 (11)	0.802
Asbestos Exposure						
No: 82	47.6 (39)	52.4 (43)		50 (41)	50 (41)	
Yes: 10	30.0 (3)	70.0 (7)	0.293	40 (4)	60 (6)	0.740
ECOG						
0–1: 86	47.7 (41)	52.3 (45)		51.2 (44)	48.8 (42)	
>2: 6	16.7 (1)	83.3 (5)	0.140	16.7 (1)	83.3 (5)	0.204
Stage						
I–III: 7	71.4 (5)	28.6 (2)		60 (6)	40. (4)	
IV: 82	43.9 (36)	56.1 (46)	0.340	47.6 (39)	52.4 (4)	0.518
Histology						
Adenocarcinoma: 79	44.3 (35)	55.7 (44)		48.1 (38)	51.9 (41)	
Squamous: 4	75.0 (3)	25.0 (1)		75 (3)	25 (1)	
Mesothelioma: 5	20.0 (1)	80.0 (4)		20 (1)	80 (4)	
Others: 4	75.0 (3)	25.0 (1)	0.245	75 (3)	25 (1)	0.276
Histological Grade						
Low: 15	46.7 (7)	53.3 (8)		53.3 (8)	46.7 (7)	
Intermediate: 27	51.9 (14)	48.1 (13)		51.9 (14)	48.1 (13)	
High: 23	34.8 (8)	65.2 (15)		34.8 (8)	65.2 (15)	
n/a: 14	42.9 (6)	57.1 (8)	0.679	57.1 (8)	42.9 (6)	0.492
Metastases						
CNS: 23	47.8 (11)	52.2 (12)	0.809	52.2 (12)	47.8 (11)	0.811
Liver: 7	42.9 (3)	57.1 (4)	0.877	42.9 (3)	57.1 (4)	1.00
Lung: 60	48.3 (29)	51.7 (31)	0.480	51.7 (31)	48.3 (29)	0.517
Ganglia: 8	62.5 (5)	37.5 (3)	0.317	62.5 (5)	37.5 (3)	0.481
Bone: 28	35.7 (10)	64.3 (18)	0.206	42.9 (12)	57.1 (16)	0.501
Pulmonary Effusion: 31	51.6 (16)	48.4 (15)	0.413	58.1 (18)	41.9 (13)	0.271
EGFR						
Wild Type: 54	44.4 (24)	55.6 (30)		46.3 (25)	53.7 (29)	
Mutant: 38	47.4 (18)	52.6 (20)	0.782	52.6 (20)	47.4 (18)	0.672
EGFR Subtype						
EXON 19: 26	42.3 (11)	57.7 (15)	0.686	46.2 (12)	47.4 (18)	
L858R: 12	50.0 (6)	50.0 (6)	0.746	58.3 (7)	41.7 (5)	
T790M: 1	100 (1)	0.0 (0)	0.273	100 (1)	0 (0)	0.547
ALK						
Wild type: 68	51.5 (35)	48.5 (33)		54.4 (37)	45.6 (31)	
Mutant: 24	29.2 (7)	70.8 (17)	0.059	33.3 (8)	66.7 (16)	0.098
TKIs Treatment						
No: 20	55.0 (11)	45.0 (9)		52.6 (20)	47.4 (18)	
Yes: 55	40.0 (22)	60.0 (33)	0.247	46.3 (25)	53.7 (29)	0.672
CTX Treatment						
No: 61	41.0 (25)	59.0 (36)		45.9 (28)	54.1 (33)	
Yes: 31	54.8 (17)	45.2 (14)	0.207	54.8 (17)	45.2 (14)	0.509
Vaccine						
BNT162b2: 31	48.4 (15)	51.6 (16)		48.4 (15)	51.6 (16)	
AZD1222: 37	45.9 (17)	54.1 (20)		48.6 (18)	51.4 (19)	
Sputnik: 10	50.0 (5)	50.0 (5)		50 (5)	50 (5)	
Sinovac: 13	38.5 (5)	61.5 (8)		46.2 (6)	53.8 (7)	
Cansino: 1	0.0 (0)	100 (1)	0.864	100 (1)	0 (0)	0.895
COVID-19						
No: 82	47.6 (39)	52.4 (43)		93.3 (42)	48.8 (40)	
Yes: 10	30.0 (3)	70.0 (7)	0.293	30 (3)	70 (7)	0.317

NSCLC, Non-Small Cell Lung Cancer. ECOG, Eastern Cooperative Oncology Group. n/a, not available. CNS, central nervous system. SARS-CoV-2, severe acute respiratory syndrome coronavirus 2. COVID-19, coronavirus disease of 2019. TKIs, tyrosine kinase inhibitors. CTX, chemotherapy. EGFR, Epidermal Growth Factor Receptor. ALK, anaplastic lymphoma kinase. AZD1222, Oxford-Astra Zeneca vaccine. BNT162b2, BioNTech-Pfizer vaccine.

**Table 3 vaccines-11-01612-t003:** SARS-CoV-2 vaccine-related adverse effects in NSCLC patients according to Common Terminology Criteria for Adverse Events (CTCAE v5.0) scale.

Clinical Characteristics N = 119	Healthy % (n)	NSCLC % (n)	*p* Value
Edema			
G0: 100 (119)	22.7 (27)	77.3 (92)	
G1≥: 0 (0)	0	0	
Erythema			
G0: 96.6 (115)	22.6 (26)	77.4 (89)	
G1≥: 3.4 (4)	25 (1)	75 (3)	0.911
Pain			
G0: 79.8 (95)	23.2 (22)	76.8 (73)	
G1≥: 20.2 (24)	20.8 (5)	79.2 (19)	0.808
Headache			
G0: 82.4 (98)	24.5 (24)	75.5 (74)	
G1≥: 17.6 (21)	14.3 (3)	85.7 (18)	0.311
Fever			
G0: 92.4 (110)	23.6 (26)	76.4 (84)	
G1: 7.6 (9)	11.1 (1)	88.9 (8)	0.388
Chills			
G0: 88.2 (105)	23.8 (25)	76.2(80)	
G1: 11.8 (14)	14.3 (2)	85.7 (12)	0.424
Diarrhea			0.440
G0: 98.3 (117)	23.1 (27)	76.9 (90)	
G1≥: 1.7 (2)	0.0 (0)	100 (2)	
Nausea and Vomiting			
G0: 95.8 (114)	22.8 (26)	77.2 (88)	
G1≥: 4.2 (5)	20 (1)	80 (4)	0.883
Fatigue			
G0: 76.5 (91)	22 (20)	78 (71)	
G1≥: 23.5 (28)	25 (7)	75 (21)	0.738
Myalgia			
G0: 86.6 (103)	25.2 (26)	74.8 (77)	
G1≥: 13.4 (16)	6.3 (1)	93.8 (15)	0.091
Arthralgias			0.694
G0: 87.4 (104)	22.1 (23)	77.9 (81)	
G1≥: 12.6 (15)	26.7 (4)	73.3 (11)	
Tachycardia			
G0: 98.3 (117)	23.1 (27)	76.9 (90)	
G1≥: 1.7 (2)	0 (0)	100 (2)	0.440
Hypertension			
G0: 100 (119)	22.7 (27)	77.3 (92)	
G1: 0 (0)	0 (0)	0 (0)	
Hypotension			
G0: 100 (119)	22.7 (27)	77.3 (92)	
G1: 0 (0)	0 (0)	0 (0)	
Induration			
G0: 99.2 (118)	22.9 (27)	77.1 (91)	
G1: 0.8 (1)	0 (0)	100 (1)	0.586

NSCLC, Non-Small Cell Lung Cancer. G0, no adverse events according to the CTCAE v5.0 grade scale. G1, mild symptoms according to the CTCAE v5.0 scale. G1≥, adverse events grade 1 of severity or higher according to the CTCAE v5.0 scale.

## Data Availability

The datasets analyzed during the current study are available from the corresponding author upon reasonable request.

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
