# Peer review of "Impact of Tyrosine Kinase Inhibitors on the Immune Response to SARS-CoV-2 Vaccination in Patients with Thoracic Malignancies"

_vaccines, 2023, doi:10.3390/vaccines11101612_

Round 1

Reviewer 1 Report

This paper attempted to uncover the Impact of Tyrosine Kinase Inhibitors on the Immune Response 2 to SARS-CoV-2 Vaccination in Patients with Non-small Cell 3 Lung Cancer. It is interesting. However, the data in the paper cannot support the author's conclusion with effect. The main concerns are listed as follow.

 1. Delete the parts in the table annotations that are duplicated with the table title

2. in the Fig 1A, the data of two sets “age<59 “and “age>59” are exactly the same, please check and correct. What is the mean “>0.9999” and “0.3666” in fig 1A and B. and the annotations “Statistical significance: p value 0.05. NSCLC, Non-Small Cell Lung Cancer” in the legend are not present in fig 1.

3. Similar issues also appear in Figure 2, 3, and 4.

4. In addition, Fig 1, 2 are too big and Fig 4 is too small. Adjust all images to be more aesthetically pleasing and to fit the text.

5. Each result section lacks a corresponding conclusion description.

6. the author claimed that “Healthy individuals developed more SARS-CoV-2 33 neutralizing antibodies than NSCLC patients……against the SARS-CoV-2 spike protein than EGFR patients.” Actually, almost all data showed insignificant statistical differences p>0.05, So, these data cannot support the author's conclusion. Maybe the author need to expand the sample size.

I think thaMinor editing of English language requiredt the amount and quality of data provided in this stu

Author Response

Thank you very much for your comments and recommendations. We improved results and conclusion.

Comment 1: Delete the parts in the table annotations that are duplicated with the table title

Response 1: We erased it.

Comment 2: in the Fig 1A, the data of two sets “age<59 “and “age>59” are exactly the same, please check and correct. What is the mean “>0.9999” and “0.3666” in fig 1A and B. and the annotations “Statistical significance: p value 0.05. NSCLC, Non-Small Cell Lung Cancer” in the legend are not present in fig 1.

Response 2: We corrected the figures and the annotations.

Comment 3: Similar issues also appear in Figure 2, 3, and 4.

Response 3: We corrected also

Comment 4: In addition, Fig 1, 2 are too big and Fig 4 is too small. Adjust all images to be more aesthetically pleasing and to fit the text.

Response 4: We adjusted the size of the  figures 

Comment 5: Each result section lacks a corresponding conclusion description.

Response 5: We added a conclusion for each result section.

Comment 6: 6. the author claimed that “Healthy individuals developed more SARS-CoV-2 33 neutralizing antibodies than NSCLC patients……against the SARS-CoV-2 spike protein than EGFR patients.” Actually, almost all data showed insignificant statistical differences p>0.05, So, these data cannot support the author's conclusion. Maybe the author need to expand the sample size.

Response 6: We modified our conclusions to align with the scope of our results

Reviewer 2 Report

The manuscript submission by Arrieta and colleagues reports on analysis of immune Response to SARS-CoV-2 vaccination in non-small cell lung cancer (NSCLC) patients. The authors collected samples from 92 patients and evaluated SARS-CoV-2 antibodies, B-cell subpopulations. They found lower antibody in patients receiving chemotherapy comparing with tyrosine kinase inhibitors and ALK patients has better immune response than EGFR patients. This is a well-conducted study and only several suggestions.

1. Line 237-241, the authors could discuss this result.

2. Line 253-263, can’t find the results.

3. Line 314, add reference.

Author Response

Thank you very much for your corrections. 

Comment 1: In lines 237-241, the authors could discuss this result.

Response 1: We added a little section in the discussion to consider this correction.

Comment 2:  Line 253-263, can’t find the results.

Response 2: We added a figure in the supplementary material to make it more visual.

Comment 3:  Line 314, add reference.

Response 3: We added this reference. Thank you for the correction.

Reviewer 3 Report

Hernández-Pedro and coauthors report the SARS-Cov-2 antibody level and B cell overall and subpopulation studies comparing healthy group vs. patients with non-small cell lung cancers upon vaccination. Blood samples were collected 30 days after the second dose of the COVID-19 vaccine to determine antibodies against SARS-CoV-2 spike protein (S protein), CD19+ B-cells, antibody-secreting B cells (ASBC), CD27 (-) B cells, and memory B lymphocytes. Participants with clinical suspicion or microbiological evidence of active COVID-19 infection were excluded during this study.

The authors concluded that antibodies against SARS-CoV-2 may be impaired in patients with NSCLC secondary to cancer treatment. Reduced antibody titers against SARS-CoV-2 and trends of lower levels of antigen- secreting cells were identified in NSCLC patients with EGFR mutations who received Tyrosine kinase inhibitors (TKI).

This study is original and contains findings that may potential benefit the Covid research and clinical community. Several comments are listed below for consideration.

(1) In the method section 2.3, it's helpful to describe the purpose of the different antibodies and their optical channel wavelengths. In addition, it's recommended to include original flow cytometry cell distribution results in figures to support the conclusions

(2) The data in Figure 2B and 3A are only described as high vs low and statistical significance but not written out or tabulated as exact average number and standard deviations. It's helpful to describe the latter too.

(3) Considering the context and scope of current studies, abstract shall draw conclusion on positive observation based on statistical significant data but clearly specify results as negative based on statistical insignificance.

Author Response

Thank you very much for your comments and corrections. We took it into account. 

Comment 1: In the method section 2.3, it's helpful to describe the purpose of the different antibodies and their optical channel wavelengths. In addition, it's recommended to include original flow cytometry cell distribution results in figures to support the conclusions

Response 1: We added a figure in the supplementary material to respond to this recommendation.

Comment 2: The data in Figure 2B and 3A are only described as high vs low and statistical significance but not written out or tabulated as exact average number and standard deviations. It's helpful to describe the latter too.

Response 2: We added the corresponding data.

Comment 3: Considering the context and scope of current studies, abstract shall draw conclusion on positive observation based on statistical significant data but clearly specify results as negative based on statistical insignificance.

Response 3: We performed some modifications in the manuscript to take into account this comment.

Reviewer 4 Report

The article submitted to Vaccines, "Impact of Tyrosine Kinase Inhibitors on the Immune Response 2 to SARS-CoV-2 Vaccination in patients with Non-small Cell 3 Lung Cancer" has significant impact in understanding the impaired response to SARS-CoV-2 vaccines in lung cancer patients who received therapy over the healthy individuals. This would be extremely important to study vaccines therapeutic effect in patients with health condition. The study did not incorporate group of patients who received third dose of vaccine which has been stated as a limiting factor and need to be addressed in future studies. This will be valuable information to further understanding SAR-CoV-2 vaccine response in diseased individuals. This article can be considered for publication after minor changes. 

Minor comments:

1. For Fig-2,3 and 4, make separate graphs for B-cell and ASC data.

2. For Fig-2,3 and 4, make separate graphs for Cd27 (-) and memory cells data.

Author Response

Thank you very much for your comments and suggestions.

Comment 1 and 2: -  For Fig-2,3 and 4, make separate graphs for B-cell and ASC data.

  • For Fig-2,3 and 4, make separate graphs for Cd27 (-) and memory cells data.

Response 1 and 2: We take into account your suggestion. We modified and improved the results sections in order to facilitate the observation and  understanding of the figure section.

Round 2

Reviewer 1 Report

In this revised version, the authour corrected many data errors and added some summaries and discussions, making the presentation of the paper clearer and more logical.  Although some concerns still exist, such as the sample size and no significant difference between some experimental groups, I have no objection to publishing these data.  

Author Response

Comment 1:  In this revised version, the author corrected many data errors and added some summaries and discussions, making the presentation of the paper clearer and more logical.  Although some concerns still exist, such as the sample size and no significant difference between some experimental groups, I have no objection to publishing these data.  

Response 1: Thank you very much for your annotations, We are aware of the potential benefits that increasing the sample size could bring. However, due to the current situation of the pandemic, recruiting patients has become more complex. We appreciate your review and corrections to our work.